# From chiral laser pulses to femto- and attosecond electronic chirality flips in achiral molecules

Yunjiao Chen[1], Dietrich Haase[2], Jörn Manz [1,2,3] ✉, Huihui Wang [1] ✉ & Yonggang Yang [1,3] ✉

Chirality is an important topic in biology, chemistry and physics. Here we show that ultrashort circularly polarized laser pulses, which are chiral, can be fired on achiral oriented molecules to induce chirality in their electronic densities, with chirality flips within femtoseconds or even attoseconds. Our results, obtained by quantum dynamics simulations, use the fact that laser pulses can break electronic symmetry while conserving nuclear symmetry. Here two laser pulses generate a superposition of three electronic eigenstates. This breaks all symmetry elements of the electronic density, making it chiral except at the periodic rare events of the chirality flips. As possible applications, we propose the combination of the electronic chirality flips with Chiral Induced Spin Selectivity.

Chirality is an important topic in femtosecond (fs) and attosecond (as) chemistry and physics[1–9]. A recent triumph is the joint experimental and theoretical discovery of chirality flips in the time-dependent part of the electronic density in a chiral molecule, specifically in methyl-lactate, with period 7.2 fs, induced by a linearly polarized UV laser pulse; this effect was monitored by a circularly polarized laser pulse[10]. It is a challenge to go two steps further, namely to apply chiral laser pulses (i) even to achiral molecules such that they generate chirality in the electronic density, with subsequent (ii) chirality flips not only in the fs but also in the as time domain. To reach this goal, our concept is to design chiral laser pulses which break all symmetry elements of the previously achiral electronic density while conserving nuclear symmetry. This is the extreme limit of previous demonstrations of partial symmetry breaking of the electronic density in the time domain up to a few fs when the nuclei stand practically still[11,12]. This effect is analogous to the effect of a gust of wind on a sailing boat, with the boat's sail, the hull and the squall playing the roles of the electrons, the nuclei and the laser pulse. During the calm before the squall, the somewhat idealized boat has $C_s$ symmetry, with the mirror plane passing through the boat's hull and sail. When the

wind blows, the (idealized elastic) sail billows, taking away the mirror plane. This makes the sail chiral, while the body of the boat still keeps $C_s$ symmetry. After the squall, the sail may flutter (i. e. flip chirality) for a while.

Here we demonstrate the effect of periodic chirality flips of the molecular electronic density on fs to as time scale induced by circularly polarized laser pulses. It is expressed by a fundamental temporal and spatial symmetry relation (7) between electronic densities with opposite chirality before and after the flips, based on Eqs. 1–6. For the derivation, let us first set the stage. The effect is demonstrated here by means of quantum dynamics simulations for the molecule NaK which has been pre-oriented by z-polarized THz and femtosecond lasers pulses;[13–15] the details are in the Supplementary Information (SI). For reference, we shall assume perfect pre-orientation even though this cannot be achieved in practice[16]. The example NaK has already served to discover other effects in the fs time domain, which were induced, however, by linearly (not circularly) polarized laser pulses[17,18]. The literature also provides the necessary highly accurate results of quantum chemistry for the electronic wavefunctions, energies and the transition dipole moments for NaK[19,20].

[1]State Key Laboratory of Quantum Optics and Quantum Optics Devices, Institute of Laser Spectroscopy, Shanxi University, Taiyuan 030006, China. [2]Institut für Chemie und Biochemie, Freie Universität Berlin, 14195 Berlin, Germany. [3]Collaborative Innovation Center of Extreme Optics, Shanxi University, Taiyuan 030006, China. ✉e-mail: jmanz@chemie.fu-berlin.de; huihuiwang2019@sxu.edu.cn; ygyang@sxu.edu.cn

## Results

### The model: NaK driven by circularly polarized laser pulses

Our application is for NaK oriented along the laboratory $z$-axis, with the Na atom pointing to positive values of $z$ (see Fig. 1). Initially, the NaK is at rest in the electronic ground state – the molecular term symbol is $1^1\Sigma^+$, but for convenience, we label it $k = 0$. Its electronic energy is set to zero, $E_{k=0} = 0$. For the electronic symmetry breaking, we apply two circularly right (+) or left (−) polarized laser pulses which propagate along the $z$-axis simultaneously. The pulses have the same Gaussian shape and the same duration, $\tau = 8$ fs, but different frequencies ($\omega_1 < \omega_2$) and different field strengths ($\varepsilon_1 < \varepsilon_2$). While transferring angular momentum, either $M\hbar = 1\hbar$ or $M\hbar = -1\hbar$, from the photons to the electrons, the laser pulses induce transitions from state $k = 0$ either to the first degenerate excited states, labeled $k = 1+$ or $k = 1-$ (molecular term symbols: $1^1\Pi_{+1}$ or $1^1\Pi_{-1}$) with energy $E_1 = \hbar\omega_1 = 2.172$ eV ($\omega_1 = 3.300$fs$^{-1}$), or to the next excited degenerate states $k = 2+$ or $k = 2-$ ($2^1\Pi_{+1}$ or $2^1\Pi_{-1}$) with energy $E_2 = \hbar\omega_2 = 2.606$ eV ($\omega_2 = 3.959$fs$^{-1}$), respectively. The corresponding state selective electronic densities $\rho_{k=0}$, $\rho_1 = \rho_{1+} = \rho_{1-}$ and $\rho_2 = \rho_{2+} = \rho_{2-}$ are also illustrated in Fig. 1, with curved arrows indicating opposite angular momenta. All state selective electronic densities $\rho_k$ adopt, of course, the cylindrical $C_{\infty v}$ symmetry of the nuclei; a detailed mathematical proof is presented in SI, specifically in Supplementary Note 2.

The laser pulses are much shorter than the vibrational periods of NaK; for reference, their harmonic values (i. e. the lower limits) are $T_{\text{vib}} = 2\pi/\omega_e = 269.5$ fs, 484.5 fs, and 402.1 fs for states $k = 0$, $1\pm$ and $2\pm$, respectively[20]. This allows the frozen nuclei approximation for the early electron dynamics, which will also be confirmed below. Specifically, the NaK bond length is frozen at equilibrium for the electronic ground state ($k = 0$), $R_e = 3.499$ Å[21,22], and the energies $E_1$ and $E_2$ are for vertical Franck-Condon type transitions[20]. Effects of moving nuclei will be discussed below.

The four possible combinations of two circularly right (+) or left (−) polarized laser pulses with frequencies $\omega_1$ and $\omega_2$ for excitations of the lower and upper $^1\Pi$ states are denoted $++$, $-+$, $+-$ and $--$, respectively. The laser fields of the individual pulses add up to the total laser field, for example, $\varepsilon_{++}(t') = \varepsilon_{1+}(t') + \varepsilon_{2+}(t')$, $\varepsilon_{-+}(t') = \varepsilon_{1-}(t') + \varepsilon_{2+}(t')$, etc. They serve to break all the electronic symmetry elements – this will be confirmed in the next sub-section.

### Generation of chiral electronic densities

The generation of chiral electronic densities is achieved by the combined circularly polarized laser pulses with electric fields $\varepsilon_{\pm\pm}(t') = \varepsilon_{++}(t')$, $\varepsilon_{-+}(t')$, etc. to excite the initial electronic eigenfunction $\Psi_{k=0}$ to superpositions of three eigenfunctions $\Psi_k$ of the states $k = 0$, $1+$ or $1-$ and $2+$ or $2-$, respectively. At the end of the laser pulses $t' = t_e$, the eigenfunctions $\Psi_k$ are multiplied by coefficients $c_{k\pm\pm}$ which depend on the $\pm\pm$ combinations of the laser pulses. They are in general complex, $c_{k\pm\pm} = d_{k\pm\pm} e^{-i\delta_{k\pm\pm}}$. Since the total wavefunctions do not depend on the overall phase, we may set $\delta_0 = 0$. The amplitudes $d_{k\pm\pm}$ are specified by the populations, $P_{k\pm\pm}(t' = t_e) = d_{k\pm\pm}^2$. The field strength $\varepsilon_1$ and $\varepsilon_2$ are tailored to excitations of 9% populations in the excited states at the end of the laser pulses ($P_k(t' = t_e) = 0.82, 0.09, 0.09$ for states $k = 0, 1\pm, 2\pm$). The laser fields and the resulting population dynamics are specified and illustrated in SI, specifically in Supplementary Note 3, cf. Supplementary Fig. 4, Fig. 1 has cartoons of the fields. All laser pulses are centered at time $t' = 0$. The corresponding maximum intensities are so weak that they do not allow any competing processes such as multi-photon excitations or ionization. The value of $t_e$ is determined by the population dynamics.

Subsequently, the wavefunctions evolve as

$$\Psi_{\pm\pm}(t) = d_{0\pm\pm}\Psi_0 + d_{1\pm\pm}\Psi_{1\pm}e^{-i(\delta_{1\pm\pm}+\omega_1 t)} + d_{2\pm\pm}\Psi_{2\pm}e^{-i(\delta_{2\pm\pm}+\omega_2 t)} \quad (1)$$

where $t$ is the time after the laser pulse ($t = 0$ at $t' = t_e$). The resulting time-dependent electronic densities in cylindrical coordinates ($r, z, \phi$) are sums of three time-independent diagonal terms plus three time-dependent off-diagonal terms. The latter depend on the right (+) or left (−) polarizations of the laser pulses,

$$\begin{aligned}\rho_{\pm\pm}(r,z,\phi,t) = &\, d_0^2\rho_0(r,z) + d_1^2\rho_1(r,z) + d_2^2\rho_2(r,z) \\ &+ 2d_0 d_1 \rho_{01\pm}(r,z)\cos(\pm\phi - \delta_1 - \omega_1 t) \\ &+ 2d_0 d_2 \rho_{02\pm}(r,z)\cos(\pm\phi - \delta_2 - \omega_2 t) \\ &+ 2d_1 d_2 \rho_{1\pm 2\pm}(r,z)\cos[\pm\phi - \delta_1 - \omega_1 t - (\pm\phi - \delta_2 - \omega_2 t)].\end{aligned} \quad (2)$$

The derivation and discussion of Eq. 2 is in SI, specifically in Supplementary Note 2. As anticipated, the electronic densities are almost always chiral, i. e. they do not have any symmetry elements

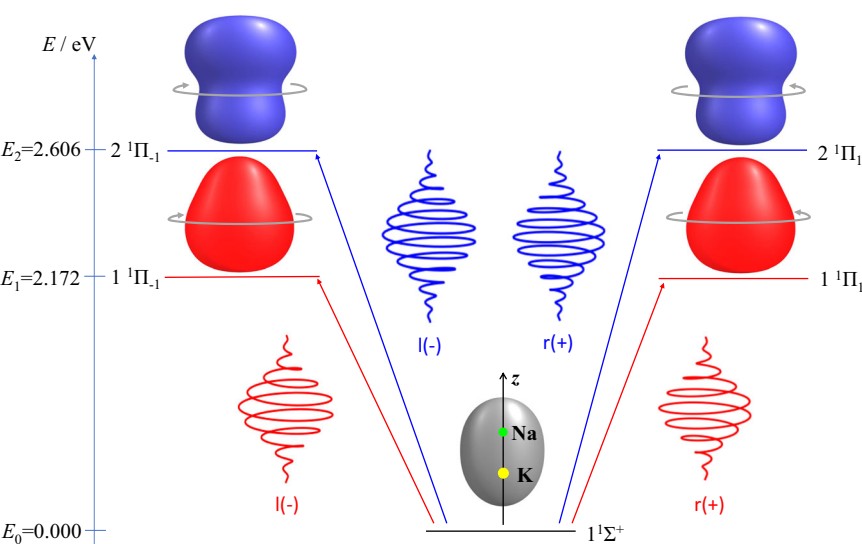

**Fig. 1 | Laser induced electronic transitions in NaK.** The NaK molecule is oriented along the laboratory $z$-axis, with the nuclei indicated by green (Na) and yellow (K) dots. The energies and the electronic densities of NaK in states $k = 0$ ($1^1\Sigma^+$), $k = 1\pm$ ($1^1\Pi_{\pm 1}$), $k = 2\pm$ ($2^1\Pi_{\pm 1}$) are also shown, for equilibrium bond length $R_e(1^1\Sigma^+) = 3.499$ Å. Circularly right (r, +, right hand side) and left (l, −, left hand side) polarized laser pulses transfer angular momenta $M\hbar = -1\hbar$ and $M\hbar = -1\hbar$ from the photons to states $1+$, $2+$ and $1-$, $2-$, respectively, indicated by curved arrows (schematic).

except for selective times $t = T_n$ where they are achiral due to a mirror plane which passes through the z-axis at special angles $\phi = \phi_n$. These exceptional events depend on two conditions which must be satisfied simultaneously, namely $\phi_n = \pm(\delta_1 + \omega_1 T_n \pm n\pi) = \pm(\delta_2 + \omega_2 T_n)$. These conditions yield two different types of solutions depending on the polarizations of the laser pulses. Equal polarizations ($+ +$ or $--$) yield

$$T_{n++} = T_{n--} = (\delta_1 - \delta_2 + n\pi)/(\omega_2 - \omega_1)$$
$$\phi_{n++} = -\phi_{n--} = (\omega_2\delta_1 - \omega_1\delta_2 + \omega_2 n\pi)/(\omega_2 - \omega_1). \qquad (3)$$

Opposite ($- +$ or $+ -$) polarizations yield

$$T_{n-+} = T_{n+-} = (-\delta_1 - \delta_2 + n\pi)/(\omega_2 + \omega_1)$$
$$\phi_{n-+} = -\phi_{n+-} = (-\omega_2\delta_1 + \omega_1\delta_2 + \omega_2 n\pi)/(\omega_2 + \omega_1). \qquad (4)$$

Obviously, the times and angles for exceptional occurrence of non-chirality of the electronic density are periodic, with periods

$$\Delta T_{++} = T_{n+1,++} - T_{n++} = \Delta T_{--} = \frac{\pi}{\omega_2 - \omega_1} = 4.764\,\text{fs}$$

$$\begin{aligned}\Delta\phi_{++} &= \phi_{n+1,++} - \phi_{n++} = -\Delta\phi_{--} \\ &= \frac{\omega_2\pi}{\omega_2 - \omega_1} = 18.862\,rad = 1080.7° = 3\times360° + 0.7°\end{aligned} \qquad (5)$$

for two laser pulses with the same ($+ +$ or $--$) circular polarization, or

$$\Delta T_{-+} = \Delta T_{+-} = \frac{\pi}{\omega_2 + \omega_1} = 0.433\,\text{fs}$$

$$\Delta\phi_{-+} = -\Delta\phi_{+-} = \frac{\omega_2\pi}{\omega_2 + \omega_1} = 1.713\,rad = 98.2° \qquad (6)$$

for two laser pulses with opposite ($- +$ or $+ -$) circular polarizations. During the entire time intervals between $T_n$ and $T_{n+1}$, the electronic densities maintain chirality.

Now consider the electronic density (2) at arbitrary position with cylindrical coordinates $(r, z, \phi = \phi_n - \gamma)$ and at arbitrary time $t = T_n - \theta$ during the period before $T_n$. Comparison of it with the electronic density at time $t = T_n + \theta$ after $T_n$, at the position reflected at the mirror plane, (i. e. at $(r, z, \phi = \phi_n + \gamma)$) yields

$$\rho_{\pm\pm}(r, z, \phi_n + \gamma, T_n + \theta) = \rho_{\pm\pm}(r, z, \phi_n - \gamma, T_n - \theta). \qquad (7)$$

The temporal and spatial symmetry relation (7) is our key result. It means that all ($\pm\pm$) combinations of two circularly right ($+$) and/or left ($-$) polarized laser pulses yield chiral electronic densities without any symmetry element at arbitrary times $T_n - \theta$ during the time interval $(T_{n-1}, T_n)$. Subsequently at time $T_n + \theta$ the electronic densities are converted to their mirror images with opposite chirality. The chirality flip occurs at $T_n$. For example, for the case of coinciding laser pulses with the same ($+ +$) circular polarizations, the snapshots of the electronic density during the time interval from $T_0$ to $T_1$ appear as sections of a left-winding double-helix, with helix pitch larger than the internuclear distance. Accordingly, we adapt IUPAC notation and assign the term "electronic $S_a$-enantiomer" during the time interval from $T_0$ to $T_1$, with subscript "a" reminding of the corresponding chiral axis which passes through the nuclei, from Na to K[23]. According to Eq. 7, this flips to the "electronic $R_a$-enantiomer" during the next time interval from $T_1$ to $T_2$, then back to the $S_a$-enantiomer from $T_2$ till $T_3$, and so on. Eq. 7 also includes the special case at $t = T_n$ when the electronic density has a mirror plane at $\phi_n$.

Another important result is that pairs of laser pulses with the same ($+ +$ or $--$) circular polarizations yield periodic chirality flips with periods in the fs time domain. In contrast, opposite ($- +$ or $+ -$) polarizations yield periodic chirality flips with periods in the as time domain (see Eqs. 5, 6). This can be rationalized by an analogy for the time evolutions of the phases in the off-diagonal terms of the electronic density (see. Eq. 2). Consider for example the "$+ +$" case of two laser pulses with the same ($+$) circular polarizations. Their effect on the phases of the time-dependent parts of the electronic densities reminds of launching a race of two runners who run in a circular stadium in the same direction. The slower and faster ones start at initial angles $\delta_1$ and $\delta_2$ and run with angular velocities $\omega_1$ and $\omega_2$, respectively. At time $T_0$ and angle $\phi_0$, they meet for the first time. For similar angular velocities (as in the present case) it takes rather long times (here: $2\Delta T_{++} = \frac{2\pi}{\omega_2 - \omega_1} = 2\times4.764$ fs) and rather long angular distance which may cost several rounds (here: $2\Delta\phi_{++} = \frac{2\omega_2\pi}{\omega_2 - \omega_1} = 2\times1080.7° = 6\times360° + 1.4°$) until they meet the next time (i. e. at $T_2$). In contrast, the "$- +$" case corresponds to two runners running in opposite directions. Clearly, in this case it costs much less time and much smaller angular distance from one meeting until the next one (here: $2\Delta T_{-+} = \frac{2\pi}{\omega_2 + \omega_1} = 2\times0.433$ fs and $2\Delta\phi_{-+} = \frac{2\omega_2\pi}{\omega_2 + \omega_1} = 2\times98.2°$).

**Illustration of ultrafast electronic chirality flips in NaK**
As an example, Fig. 2 illustrates the effects of two circularly polarized laser pulses with the same ($+ +$, Fig. 2a) and with opposite ($- +$, Fig. 2b) polarizations on the electronic density of oriented NaK. The times $T_n$ of the chirality flips with mirror planes at angles $\phi_n$ are listed in Table 1. The laser parameters are the same for the two cases, but the $+ +$ and $- +$ combinations of the circular polarizations yield entirely different chirality flips of the electronic densities, with periods $\Delta T_{++} = 4.764$ fs and $\Delta T_{-+} = 0.433$ fs, respectively.

The snapshots illustrate the time-dependent parts of the electronic density (2). They exemplify the general rule (7), namely that the electronic densities are almost always chiral, except at special periodic times $T_n$ ($n = 0, 1, 2, ...$) when the chirality flips from electronic $S_a$- to $R_a$- enantiomers, and back. The snapshots of the chiral densities at time $T_n + \theta$ are mirror images of the previous densities with opposite chirality at time $T_n - \theta$, with respect to the mirror plane at $T_n$. Additional snapshots are in SI, specifically in Supplementary Note 1, cf. Supplementary Fig. 1–3. They document charge migration; for a recent survey of different linear, planar or three-dimensional (3D) paths of the center of charge during charge migration, see (Ref. 24,25). The fact that the present electronic densities are almost always chiral (except at the instants of chirality flips) does not imply, however, that the path of charge migration is also chiral, e.g. helical. Electronic chirality flips and helical charge migration[25] are two different and independent phenomena which may be induced by different chiral laser pulses.

The present results are for laser driven coherent electron dynamics based on the approximation of frozen nuclei. This is well justified for the initial time domain of the generation of electronic chirality and for the first chirality flips, in particular when they occur in the as time domain, which is about two orders of magnitude shorter than the periods of nuclear motion. At longer times, however, the coupling of the electrons to the nuclei will cause decoherence[26]. The present system has exceptionally long decoherence time (compared to hundreds of others[27]), $\tau_{dec} > 20$ fs (estimated in Supplementary Note 4 of SI by means of the methods of (Refs. 28–32)). This confirms the validity of the approximation of frozen nuclei for the present application.

## Discussion
We have discovered that chirality in the electronic density of an achiral molecule can be generated by means of two circularly polarized laser pulses with the same or with opposite polarizations, followed by periodic chirality flips in the fs or as time domains, respectively. The electronic chirality can be characterized by Grimme's Continuous Chirality Measure (CCM)[33]. Accordingly, the CCM is equal to zero at the

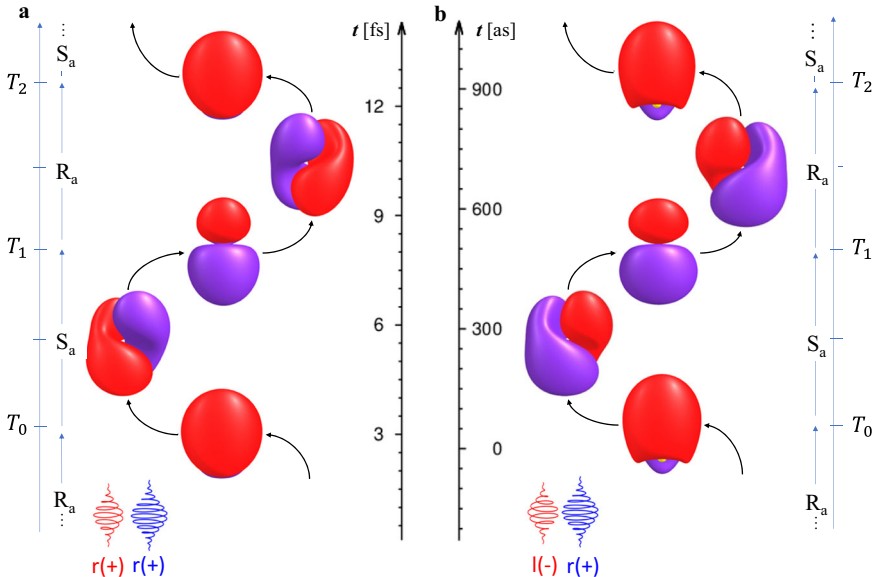

**Fig. 2 | Chirality flips between electronic $S_a$- and $R_a$-enantiomers of oriented NaK.** The chirality flips with periods in the femtosecond (**a**) and attosecond (**b**) time domains are induced by two circularly polarized laser pulses (schematic) with the same (**a**, + +) and with opposite (**b**, − +) polarizations, respectively, cf. Fig. 1. The laser pulses for the cases (**a**) and (**b**) have the same parameters, see supplementary Table 1. The snapshots show the positive (red) and negative (blue) time-dependent parts of the electronic densities. They are almost always chiral, except at the periodic instants labeled $T_{n++}$ or $T_{n-+}$ ($n = 0,1,2 \ldots$) when they have mirror planes at $\phi_{n++}$ or $\phi_{n-+}$ for chirality flips from $S_a$ to $R_a$ or back from $R_a$ to $S_a$, cf. Table 1.

**Table 1 | Times and mirror planes for chirality flips of the electronic densities in NaK[a]**

| $n$ | $T_{n++} = T_{n--}$ (fs) | $\phi_{n++} = -\phi_{n--}$ (°) | $T_{n+-} = T_{n-+}$ (as) | $\phi_{n+-} = -\phi_{n-+}$ (°) |
|---|---|---|---|---|
| 0 | 3.012 | 261.3 | 18.9 | 302.6 |
| 1 | 7.775 | 262.0 | 451.6 | 40.8 |
| 2 | 12.539 | 262.7 | 884.4 | 139.0 |

[a]The chirality flips are induced by two circularly polarized laser pulses with the same (+ +, −−) or with opposite (+ −, − +) polarizations. At times $T_n$, mirror planes pass through the z-axis at angle $\phi_n$, cf. Eqs. 3–6. For the cases + + or −−, the event labeled $n = 2$ is followed by events $n = 3,4,5,\ldots$ with periods $\Delta T = 4.764$ fs, $\Delta\phi = 1080.7° = 3 \times 360° + 0.7°$. For the cases + − or − +, the event labeled $n = 2$ is followed by events $n = 3,4,5,\ldots$ with periods $\Delta T = 0.433$ fs, $\Delta\phi = 98.2°$.

times $T_n$ of the chirality flips when the electronic chirality is achiral, and it achieves its maximum value at times $T_n + \Delta T/2 = (T_n + T_{n+1})/2$ half way between the chirality flips at $T_n$ and $T_{n+1}$. Details will be published elsewhere.

For achiral molecules such as NaK, the laser pulses must break all symmetry elements of the electronic density, making it chiral while the nuclear structure remains achiral. For comparison, the electronic density of chiral molecules is already chiral from the outset. Hence it is not necessary to break any symmetry elements so that a single linearly polarized laser pulse suffices to induce the chirality flips[10].

The present chirality of the electronic density is generated in a superposition state of three electronic eigenstates of the oriented NaK, namely the ground state and two excited states which have the same irreducible representations (here: $\Pi$) different from the ground state ($\Sigma^+$) (see Eq. 1). The superposition of three eigenstates is unique. For comparison, the superposition of two eigenstates could not break all the symmetry elements of $C_{\infty v}$, which means the electronic density would remain achiral. Superpositions of more than three eigenstates with incommensurable frequencies would not allow periodic electron dynamics. The present superposition of three molecular electronic eigenstates is reminiscent of the superposition of three atomic orbitals representing the chiral hydrogen atom[7,34]. The phenomenon of periodic chirality flips is of course not restricted to the electronic density of oriented heteronuclear diatomic molecules. It should also exist in any other achiral polyatomic linear molecules with $C_{\infty v}$ symmetry. Extensions to symmetric linear molecules with $D_{\infty h}$ symmetry would call for an additional (presumably linearly z-polarized) laser pulse in order to

take away the horizontal symmetry plane. Extensions to nonlinear achiral molecules are also possible, again by use of laser pulses that break all symmetries of the nuclear frame in the electronic density.

Recent quantum dynamics simulations of the imaging of charge migration in chiral molecules suggest that the electronic chirality flips can be measured by time-resolved x-ray diffraction[35]. The differential scattering probabilities (DSP) are calculated by consistent quantum electrodynamics description of light and quantum mechanical description of the electron dynamics[36]. Different DSPs for opposite electronic enantiomers imply that the present periodic electronic chirality flips should yield periodically alternating DSPs. The required ultrashort x-ray pulses are available, cf. Ref. 37. Promising complementary techniques include chiral spectroscopy by dynamical symmetry breaking in high harmonic generation[6], ultrafast imaging of chiral dynamics by non-linear enantio-sensitive optical response to synthetic chiral light[38], ultrafast photoelectron imaging of attosecond electron dynamics[5,39–42], or photoelectron circular dichroism (PECD)[7,10,34,43]. As a caveat, the reconstruction of three-dimensional time-dependent electronic densities by means of the images obtained via photoionization, such as PECD, requires the elimination of the contributions caused by the electric fields, or their vector potentials, which induce photoionization. Nevertheless, time-resolved variations of the images of alternating enantiomers could provide valuable quantitative information about the periods of the chirality flips and about their equivalence in the femtosecond and attosecond time domains, as generated by coinciding laser pulses with the same or with opposite circular polarizations, respectively.

As possible application, we propose to combine the present electronic chirality flips with the effect of chiral induced spin selectivity (CISS). CISS means that electrons which flow through a chiral molecule become spin-polarized[44–46]. In spite of many efforts (see e. g. the recent special issue dedicated to research related to the CISS effect[47]), according to (Ref. 48) "the detailed underlying mechanism is still an open question". We assume that CISS is due to the interactions of spin-selective electrons passing through chiral electronic densities. The present chirality flips should then modulate the CISS signals on the fs to as time scale, analogous to the switches of the direction of spin-polarization induced by chirality flips during the rotation of a molecular motor[49]. The effects will be reduced by finite orientation (imperfect axis distribution)[13–16,50,51]. By extrapolation of the abstract of (Ref. 48), this should have implications for various applications e.g. spintronics, adding another twist to the role of controlling electrons in attochemistry[52].

This work establishes the generation of electronic chirality flips in achiral molecules by means of two circularly polarized laser pulses with opposite ($+ -$ or $- +$) polarizations as a new topic in Attosecond Chemistry and Physics. Our quantum dynamics simulations provide a short-time record for the conversion of (here: electronic) enantiomers, opposite to the long-time stability of biochemical enantiomers, which may last for millions of years[53–55].

## Methods

The potential energy curves, transition dipoles, and the corresponding electronic wavefunctions of NaK are calculated by the state-averaged CASSCF[56] method as implemented in MOLPRO[57]. The basis functions are the same as the ones optimized in (Ref. 20) for NaK. Subsequently, we optimized the laser parameters by quantum dynamics simulations of five states (cf. Fig. 1) with fixed internuclear distance $R = R_e$. Details are in SI, specifically in Supplementary Note 3, cf. Supplementary Figure 4. The same laser parameters are used for four different combinations of polarizations to obtain the target expansion coefficients. The time-dependent electronic density is calculated as the mean value of the density operator. The final expression (2) is derived in SI, specifically in Supplementary Note 2. It depends on the expansion coefficients of the involved electronic wavefunctions and the transition densities. The expansion coefficients are determined and listed in Supplementary Note 3 of SI, cf. Supplementary Table 1. The transition densities are calculated by means of the ORBKIT package[58,59].

Nuclear motions of the NaK molecule lead to decoherences of electron dynamics. However, the point-group symmetry of the molecule is conserved. The decoherence time of the target superposition states is estimated by full quantum dynamics simulations of the time evolutions of nuclear Franck-Condon wavepackets on individual potential energy curves, leading to a coherence time larger than 20 fs. We further performed full quantum dynamics simulations, using the laser parameters for the preparation of the target superposition states, assuming the molecule is in the electronic and vibrational ground state before the laser pulses. The overlaps between the involved nuclear wavepackets remain close to one (which means very high degrees of electronic coherence[28–32]) at the end of the laser pulses, as documented in SI, specifically in Supplementary Note 4, cf. Supplementary Figs. 5, 6. For all the quantum dynamics simulations with nuclear motions, the split operator method[60] is used for the numerical propagations of the wavepackets.

## Data availability

The data that are necessary to interpret, verify and extend the research in the article are provided in the main text and/or SI. All the raw data files are available from the corresponding authors upon request.

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

## Acknowledgements

We are very grateful to Dr. ChunMei Liu (Nanjing University of Posts and Telecommunications) for stimulating discussions, to Dr. P. Jasik (Gdańsk University of Technology) for providing the input file for the highly accurate quantum chemical calculations of the electronic energies and wavefunctions of NaK[20] and to Prof. D. J. Diestler (University of Nebraska-Lincoln) for advice on the manuscript. One of us (J.M.) also thanks Dr. S. Naskar (Universität Hamburg and The Hamburg Centre for Ultrafast Imaging) for pointing to possible applications of electronic chirality flips to the CISS effect[48], to Dr. D. Ayuso (Max Born Institute, Berlin and

Imperial College, London) and to Prof. J. C. Tremblay (Université Lorraine, Metz) for illuminating discussions, and also to Professor A. D. Bandrauk (Université de Sherbrooke), to PD Dr. M. Leibscher (Freie Universität Berlin) and to PD Dr. B. Schmidt (Weierstraß-Institut, Berlin) for pointing to important references. Two of us (D.H. and J. M.) thank Prof. Beate Paulus (Freie Universität Berlin) for continuous support. This work was supported in part by the National Key Research and Development Program of China (Grants No. 2022YFA1404201 (H. W.)), the National Natural Science Foundation of China (Grants No. 12374267 (Y. Y.), U22A2091 (Y. Y.), 11904215 (H. W.)), the 111 Project (Grant No. D18001 (J. M.)), the Fund for Shanxi 1331 Project Key Subjects Construction (Y. Y.), and the Hundred Talent Program of Shanxi Province (J. M.).

## Author contributions

J.M. provided the concept. D.H. and J.M. suggested the method. Y.C., H.W. and Y.Y. obtained all the results and prepared all the figures. J.M. wrote the original draft. All authors contributed to the final version of the manuscript.

## Competing interests

Authors declare no competing interests.
