## [Peer Review File · Nature Communications]

From chiral laser pulses to femto- and attosecond electronic chirality flips in achiral moleculesREVIEWER COMMENTS

Reviewer #1 (Remarks to the Author):

Chirality of molecules is one of the most essential concepts in nature. So far the concept is used for enantiomers in the stationary state, while in the manuscript the authors extended “chirality” to molecules in nonstationary state. They found that the electron densities of achiral molecules, excited by two laser pulses with circularly polarizations, have chirality behaviors, that is, chirality flips with periods in the fs and as time domain. The two types of the chirality flips come from the same or different circularly polarizations of the two pulses. It is very interesting that the flipping time depends on the circularly polarizations of the two pulses. This was clarified in two ways, one was by analytically solving the time-dependent Schrödinger equations, and the other by performing quantum chemical dynamic simulations. A hetero diatomic molecule, NaK, was taken as a simple example. The details were clearly described in Supplemental information.

Studies of ultrafast nonstationary behaviors of electrons in molecules are important research targets. As described in the paper, control of the electrons in molecules by two laser pulses with helical polarizations is the first step for as switching devices in the next generation.

I am sure that the paper will make a significant contribution to development of new research fields of ultrafast chiral behaviors of the electrons in achiral molecules and application to as control of electrons in molecules. In conclusion the manuscript is suitable for publication to “Nature Communications” after the authors consider questions described below.

Questions:

Q1. Analytical expressions for the periods in fs and as time domain are given in Eqs. (4) and (6), respectively. The origin of the periods in fs is the coherence between the two excited states, which is independent of the laser polarizations. The origin of the flipping in as time domain, on the other hand, depends on the laser polarizations of the two pulses. Is it possible to give an appropriate explanation of Eq. (6), why the flipping time is inversely proportional to the sum of the frequencies of the two pulses?

Q2. The authors introduced electronic R-enantiomer, electronic S-enantiomer for the nonstationary electron densities, using S and R convention, similar to the nomenclature for chiral compounds in the stationary state. They tentatively called the electron density during the time interval from T0 to T1 the “electronic R-enantiomer” and the flipped electron density during the time interval from T1 to T2 the “electronic S-enantiomer”, which should be relevant to two circularly right(+) and left(-) polarized laser pulses. Therefore, I think that the newly introduced nomenclature is inadequate.

Reviewer #2 (Remarks to the Author):

The authors present an idea to create chiral states in oriented achiral molecules using circularly polarized fields with one and two frequencies. They demonstrate this idea by performing quantum dynamics simulations in the heteronuclear diatomic molecule NaK, and show that the chirality of the electronic excited states changes on a femtosecond time scale. I find the paper very interesting and well written, but I do not think that the findings of this work are sufficiently impactful for publication in Nature Communications. Therefore, I am afraid that I have to recommend that the paper is rejected, and that the authors consider publication in a more specialised journal, such as Communications Physics.

I would recommend that the authors considering these two points, either for this or for future publications:

1. It would be interesting to describe and quantify (via quantum dynamic simulations) how this chirality could be measured in an experiment.
2. It would also be interesting to present a way of characterising this electronic chirality.

Reviewer #3 (Remarks to the Author):

This manuscript describes a pioneering theoretical study of the creation of electronic chirality in heteronuclear diatomic molecules. Using accurate electronic-structure calculations of the NaK molecule, the authors study the time-dependent electronic chirality of superposition states of the (Σ) ground state and the two lowest-lying (Π) electronically excited states created by two resonant laser pulses. The authors show that co-rotating laser pulses create a time-dependent chiral electron density that flips chirality with a period of 4.804 fs, whereas counter-rotating pulses create chirality that flips with a period of 0.433 fs.

This is a truly visionary piece of work, which addresses the fascinating topic of chirality on an electronic time scale and demonstrates how electronic chirality can be created and how it evolves on attosecond and few-femtosecond time scales. Another beauty of the present manuscript are the classical analogies that are given, such as the sail that breaks the C_s symmetry of the sailing boat under the effect of the wind or the runners in the stadium that run in the same or opposite directions.

The work was obviously carried out with great care, is well described and very clearly presented. The electronic-structure calculations are state of the art, the derivations are given with sufficient detail that they can easily be followed and the illustrations are of highest quality. The work is also nicely placed in the contemporary context of the attosecond literature. The outlooks are promising and highlight the potentially fundamental implications and the significance of time-dependent electronic chirality in molecules.

Overall, I strongly recommend acceptance of this beautiful piece of work. The authors may consider adding some discussions of the following aspects:

- 1) The authors assume the molecules to be perfectly oriented, which cannot be achieved in practice. Could the authors comment on the impact of finite orientation (imperfect axis distribution) on a) the possibility to observe the electronic chirality and b) how finite orientation impacts any of the reaction-dynamics effects that could be induced by the electronic chirality.
- 2) The authors mention photoelectron circular dichroism as a possible way to detect the electronic chirality. However, as the authors know, PECD is often dominated by the chirality of the scattering potential, rather than the chirality of the electronic wavefunction. Comments on this aspect would be highly welcome.
- 3) The authors make the point that a single circularly polarized laser pulse would not be capable of inducing the electronic chirality. I am not sure that I understand this aspect because a single ultrashort broadband pulse that covers the energies of both excited states and a controllable spectral phase should be capable of mimicking the effect of the two 8-fs pulses with different frequencies and phases. Comments on this aspect would also improve the manuscript further.

In summary, I strongly recommend this visionary work for publication in Nature Communications, with optional revisions as described above.

Response to the reviewers' comments

Reviewer #1 (Remarks to the Author):

Chirality of molecules is one of the most essential concepts in nature. So far the concept is used for enantiomers in the stationary state, while in the manuscript the authors extended “chirality” to molecules in nonstationary state. They found that the electron densities of achiral molecules, excited by two laser pulses with circularly polarizations, have chirality behaviors, that is, chirality flips with periods in the fs and as time domain. The two types of the chirality flips come from the same or different circularly polarizations of the two pulses. It is very interesting that the flipping time depends on the circularly polarizations of the two pulses. This was clarified in two ways, one was by analytically solving the time-dependent Schrödinger equations, and the other by performing quantum chemical dynamic simulations. A hetero diatomic molecule, NaK, was taken as a simple example. The details were clearly described in Supplemental information. Studies of ultrafast nonstationary behaviors of electrons in molecules are important research targets. As described in the paper, control of the electrons in molecules by two laser pulses with helical polarizations is the first step for as switching devices in the next generation.

I am sure that the paper will make a significant contribution to development of new research fields of ultrafast chiral behaviors of the electrons in achiral molecules and application to as control of electrons in molecules. In conclusion the manuscript is suitable for publication to “Nature Communications” after the authors consider questions described below.

Our response: We are grateful to Reviewer #1 for investing time and expertise in reviewing our manuscript. We are glad about the positive evaluation.

Questions:

Q1. Analytical expressions for the periods in fs and as time domain are given in Eqs. (4) and (6), respectively. The origin of the periods in fs is the coherence between the two excited states, which is independent of the laser polarizations. The origin of the flipping in as time domain, on the other hand, depends on the laser polarizations of the two pulses. Is it possible to give an appropriate explanation of Eq. (6), why the flipping time is inversely proportional to the sum of the frequencies of the two pulses?

Our response: The result, Eq. (6), can be rationalized by the analogy for the time evolutions of the phases in the first and second off-diagonal terms (see Eq. (2)) of the electron density with two runners who run in a circular stadium, see the second paragraph after Eq. (7). Opposite polarizations of the laser pulses launch the race of the two runners in opposite directions. Let the period for meetings be $2 \times \Delta T$. During this period, the angular positions (phases) of the first and second runners are $\phi_1 = \omega_1 \Delta T$ and $\phi_2 = \omega_2 \Delta T$, respectively. At the first meeting, the angular positions (phases) add up to $\phi_1 + \phi_2 = (\omega_1 + \omega_2) 2\Delta T = 2\pi$, hence $\Delta T = \pi/(\omega_1 + \omega_2)$, as in Eq. (6). In contrast, equal polarizations launch the race of the two runners in the same direction. At the first meeting, the angular position of the faster runner exceeds the slower runner by 2π . This means $\phi_2 - \phi_1 = (\omega_2 - \omega_1) 2\Delta T = 2\pi$, hence $\Delta T = \pi/(\omega_2 - \omega_1)$, cf. Eq. (5).

Modification of the text: In the last line of the second paragraph after Eq. (7), we have added “ $2 \Delta T_{+} = 2\pi/(\omega_1 + \omega_2)$, ..., cf. Eq. (6)”. In line 5 from bottom of the second paragraph after Eq. (7), we have added “ $2 \Delta T_{++} = 2\pi/(\omega_2 - \omega_1)$, ..., cf. Eq. (5)”.

Q2. The authors introduced electronic R-enantiomer, electronic S-enantiomer for the nonstationary electron densities, using S and R convention, similar to the nomenclature for chiral compounds in the stationary state. They tentatively called the electron density during the time interval from T_0 to T_1 the “electronic R-enantiomer” and the flipped electron density during the time interval from T_1 to T_2 the “electronic S-enantiomer”, which should be relevant to two circularly right(+) and left(-) polarized laser pulses. Therefore, I think that the newly introduced nomenclature is inadequate.

Our response: We thank Reviewer #1 for questioning our tentative assignment of “electronic R- and S-enantiomers”. To be on the safe side, we have checked and adapted IUPAC notation. Accordingly, the previous wrong assignments R and S are replaced by S_a and R_a , respectively, with subscript “a” which reminds of the chiral axis which passes through the nuclei, from Na to K. (A hint: IUPAC would also allow the alternative notation M and P, respectively.) In fact, the snapshots of the electron density for S_a and R_a enantiomers remind of sections of left (“sinister”, S_a) - and right (“rectus”, R_a) - winding double-helices with pitches larger than the internuclear distance.

Modifications of the text: We have replaced the previous sentence in lines 5-8 after Eq. (7) (“Since there is no established nomenclature...the “electronic R-enantiomer””) by the following: “For example, for the case of coinciding laser pulses with the same (++) circular polarizations, the snapshots of the electron density during the time interval from T_0 to T_1 appear as sections of a left-winding double-helix, with helix pitch larger than the internuclear distance. Accordingly, we adapt IUPAC notation and assign the term “electronic S_a -enantiomer” during the time interval from T_0 to T_1 , with subscript “a” reminding of the corresponding chiral axis which passes through the nuclei, from Na to K²³. ”

The newly added Reference [23] is: “IUPAC-Compendium of Chemical Terminology, 2nd edition (the “Gold Book”), compiled by A. D. McNaught and A. Wilkinson, Blackwell Sci. Publ., Oxford (1997). <https://doi.org/10.1351/goldbook.A00547>”

Subsequently, we have replaced all the previous notations for “R- and S-enantiomers” systematically by “ S_a - and R_a -enantiomers” both in the text and in the Figures and Figure legends.

Reviewer #2 (Remarks to the Author):

The authors present an idea to create chiral states in oriented achiral molecules using circularly polarized fields with one and two frequencies. They demonstrate this idea by performing quantum dynamics simulations in the heteronuclear diatomic molecule NaK, and show that the chirality of the electronic excited states changes on a femtosecond time scale. I find the paper very interesting and well written, but I do not think that the findings of this work are sufficiently impactful for publication in Nature Communications. Therefore, I am afraid that I have to recommend that the paper is rejected, and that the authors consider publication in a more specialised journal, such as Communications Physics.

Our response: We are grateful to Reviewer #2 for investing time and expertise in reviewing our manuscript. But we are a bit disappointed since Reviewer #2 calls special attention to our first results which demonstrate that the chirality of the electronic excited states changes on the femtosecond time scale, but ignores our second results which discover the effect on the attosecond time scale. These second results provide a break-through for the phenomenon of electronic chirality flips in achiral molecules, from Femtosecond Chemistry to Attosecond Chemistry. The importance of electron dynamics in the attosecond time domain is confirmed by the 2023 Nobel Prize in Physics.

I would recommend that the authors considering these two points, either for this or for future publications:

1. It would be interesting to describe and quantify (via quantum dynamic simulations) how this chirality could be measured in an experiment.

Our response: Recent quantum dynamics simulations of the imaging of charge migration in chiral molecules suggest that the electronic chirality flips can be measured by time-resolved x-ray diffraction³⁸. The differential scattering probabilities (DSP) are calculated by consistent quantum electrodynamics description of light and quantum mechanical description of the electron dynamics³⁹. Different DSPs for opposite electronic enantiomers imply that the present periodic electronic chirality flips should yield periodically alternating DSPs. The required ultrashort x-ray pulses are available, cf. Ref. 40. Promising complementary techniques include ultrafast chiral spectroscopy by dynamical symmetry breaking in high harmonic generation⁶, ultrafast imaging of chiral dynamics by non-linear enantio-sensitive optical response to synthetic chiral light⁴¹, ultrafast photoelectron imaging of attosecond electron dynamics^{5,42-45}, or photoelectron circular dichroism.^{7,10,37,46}

The newly added References [38-46] are:

38. Giri, S., Tremblay, J. C., & Dixit, G. Imaging charge migration in chiral molecules using time-resolved x-ray diffraction. *Phys. Rev. A* **104**, 053115 (2021).
39. Dixit, G., Vendrell, O. & Santra, R. Imaging electronic quantum motion with light. *PNAS* **109**, 11636 (2012).
40. Hermann, G., Pohl, V., Dixit, G. & Tremblay, J. C. Probing Electronic Fluxes via Time-resolved X-Ray Scattering, *Phys. Rev. Lett.* **124**, 013002 (2020).
41. Ayuso, D., Neufeld, O., Ordonez, A. F., Decleva, P., Lerner, G., Cohen, O., Ivanov, M., Smirnova, O. Synthetic chiral light for efficient control of chiral light-matter interaction. *Nat. Photon.* **13**, 866-871 (2019).
42. Mignolet, B., Levine, R. D. & Remacle, F. Localized electron dynamics in attosecond-pulse-excited molecular systems: Probing the time-dependent electron density by sudden photoionization. *Phys. Rev. A* **86**, 053429 (2012).
43. Yuan, K.-J. & Bandrauk, A. D. Time Resolved Photoelectron Imaging of Molecular Coherent Excitation and Charge Migration by Ultrashort Laser Pulses. *J. Phys. Chem. A* **122**, 2241-2249 (2018).
44. Yuan, K.-J. & Bandrauk, A. D. Ultrafast X-ray Photoelectron Imaging of Attosecond Electron Dynamics in Molecular Coherent Excitation. *J. Phys. Chem. A* **123**, 1328-1336 (2019).
45. Reuner, M. & Popova-Gorelova, D. Attosecond imaging of photoinduced dynamics in

molecules using time-resolved photoelectron momentum microscopy. *Phys. Rev. A* **107**, 023101 (2023).

46. Ordóñez, A. F. & Smirnova, O. Propensity rules in photoelectron circular dichroism in chiral molecules. II. General picture. *Phys. Rev. A* **99**, 043417 (2019).

Modifications of the text: We have inserted the following new paragraph (with additional References [38-46]) between the previous second last and the last paragraphs:

“... in the electronic density.

Recent quantum dynamics simulations of the imaging of charge migration in chiral molecules suggest that the electronic chirality flips can be measured by time-resolved x-ray diffraction³⁸. The differential scattering probabilities (DSP) are calculated by consistent quantum electrodynamics description of light and quantum mechanical description of the electron dynamics³⁹. Different DSPs for opposite electronic enantiomers imply that the present periodic electronic chirality flips should yield periodically alternating DSPs. The required ultrashort x-ray pulses are available, cf. Ref. 40. Promising complementary techniques include chiral spectroscopy by dynamical symmetry breaking in high harmonic generation⁶, ultrafast imaging of chiral dynamics by non-linear enantio-sensitive optical response to synthetic chiral light⁴¹, ultrafast photoelectron imaging of attosecond electron dynamics^{5,42-45}, or photoelectron circular dichroism^{7,10,37,46}.

Possible applications of the present chirality flips ...”

See also the extended modification of the text, in reply to the related item Q2 of Reviewer #3.

2. It would also be interesting to present a way of characterising this electronic chirality.

Our response: We are grateful to Reviewer #2 for raising this second point since it stimulates investigations in an up-coming sub-field of Femtosecond and Attosecond Chemistry, namely the characterization of electronic chirality on ultrafast time scale. The present rapid flips between opposite enantiomers imply that the electronic density is achiral at the instants T_n of the flips. Hence, the “degree of chirality” should be equal to 0 at T_n , and then it should grow, reach a maximum and vanish as time increases from T_n to the next chirality flip at T_{n+1} . Recently, N. Mayer et al⁴¹ introduced what they call “an unambiguous chiral measure”. Essentially, it evaluates the difference between the (normalized) photoelectron distributions of opposite enantiomers. In view of Reviewer #3’s item (2), this measure might suffer, however, from the same shortcomings as for photoelectron circular dichroism, namely it might be “dominated by the chirality of the scattering potential, rather than the chirality of the electronic wavefunction”. To avoid this problem, we prefer it to characterize electronic chirality in terms of the electronic wavefunction ψ directly, without the detour via imaging by photoelectrons or other means, e. g. by X-rays. For this reason, we suggest to adapt the “Continuous symmetry measure of electronic wave functions”, as developed and applied to chiral molecules by S. Grimme, *Chem. Phys. Lett.* **297**, 15 (1998). Essentially, Grimme’s “Continuous chirality measure” (CCM) is $CCM = 100 * \min(1 - |\langle \psi | \hat{S} | \psi \rangle|)$ where “100” is a scaling factor for expressing chirality in percent (%), and the minimum is to be taken with respect to the symmetry operators \hat{S} which destroy chirality, that means $\hat{S}|\psi\rangle = \pm|\psi\rangle$ for the reference case of achiral electronic states. In that case, $CCM = 0$, else $CCM > 0$, with the limit $CCM = 100\%$ for

maximum chirality. Originally, Grimme developed his CCM for non-degenerate (time-independent!) electronic eigenstates. We introduce two extensions, namely (i) to time-dependent electronic wave functions $\psi(t)$ which are superpositions of non-degenerate and (ii) degenerate eigenstates. Our fundamental symmetry relation, Eq. (7) suggests the symmetry operators \hat{s} . As an example, the resulting $CCM(t) = 100 * \min(1 - |\langle \psi(t) | \hat{s} | \psi(t) \rangle|)$ for the electronic wavefunction driven by two right circularly polarized laser pulses is shown here, for the time domain from $T_1 - \Delta T/2$ to $T_1 + \Delta T/2$, where T_1 is the time of the first chirality flip, with period ΔT .

Figure for $CCM(t)$ versus time t

Our extension of Grimme's CCM to (i) time-dependent superpositions of non-degenerate and (ii) also degenerate electronic eigenstates, together with the new applications to electronic chirality flips in the femto- and attosecond time domains provide significant progress of the field. We prefer it to present the details in an independent sequel publication.

Modifications of the manuscript: At the end of the first paragraph of the conclusions, we have added:

“The electronic chirality can be characterized by Grimme's Continuous Chirality Measure (CCM)³⁶. Accordingly, the CCM is equal to zero at the times T_n of the chirality flips when the electronic chirality is achiral, and it achieves its maximum value at times $T_n + \Delta T/2 = (T_n + T_{n+1})/2$ half way between the chirality flips at T_n and T_{n+1} . Details will be published elsewhere.”

The newly added Reference [36] is: Grimme, S. Continuous symmetry measures for electronic

wave functions, *Chem. Phys. Lett.* **297**, 15-22 (1998).

Reviewer #3 (Remarks to the Author):

This manuscript describes a pioneering theoretical study of the creation of electronic chirality in heteronuclear diatomic molecules. Using accurate electronic-structure calculations of the NaK molecule, the authors study the time-dependent electronic chirality of superposition states of the (Σ) ground state and the two lowest-lying (Π) electronically excited states created by two resonant laser pulses. The authors show that co-rotating laser pulses create a time-dependent chiral electron density that flips chirality with a period of 4.804 fs, whereas counter-rotating pulses create chirality that flips with a period of 0.433 fs.

This is a truly visionary piece of work, which addresses the fascinating topic of chirality on an electronic time scale and demonstrates how electronic chirality can be created and how it evolves on attosecond and few-femtosecond time scales. Another beauty of the present manuscript are the classical analogies that are given, such as the sail that breaks the C_s symmetry of the sailing boat under the effect of the wind or the runners in the stadium that run in the same or opposite directions.

The work was obviously carried out with great care, is well described and very clearly presented. The electronic-structure calculations are state of the art, the derivations are given with sufficient detail that they can easily be followed and the illustrations are of highest quality. The work is also nicely placed in the contemporary context of the attosecond literature. The outlooks are promising and highlight the potentially fundamental implications and the significance of time-dependent electronic chirality in molecules.

Overall, I strongly recommend acceptance of this beautiful piece of work.

Our response: We are grateful to Reviewer #3 for investing time and expertise in reviewing our manuscript. We are glad about the positive evaluation.

The authors may consider adding some discussions of the following aspects:

- 1) The authors assume the molecules to be perfectly oriented, which cannot be achieved in practice. Could the authors comment on the impact of finite orientation (imperfect axis distribution) on a) the possibility to observe the electronic chirality and b) how finite orientation impacts any of the reaction-dynamics effects that could be induced by the electronic chirality.

Our response: We confirm that our model assumption of perfect orientation is an approximation which cannot be achieved in practice¹⁶. Nevertheless it is gratifying that one can achieve rather high values (e. g., the mean value $\langle \cos \theta \rangle$ larger than 0.7¹⁴ or even 0.8^{13,15}) of axial orientations, and even more importantly, these axial distributions can be maintained field-free for much longer times (e. g. >100 fs¹³, >250 fs¹⁴ or >500 fs¹⁵) than the durations of the present laser pulses and the subsequent electronic chirality flips.

For the requested estimate of the impact of finite axial orientation angles $0 < \theta < \pi$ (with typical

value $\cos \theta = 0.7$) and $0 \leq \varphi < 2\pi$, following the example of Ref. 38, we have carried out new quantum dynamics simulations for the preparation of the $1^1\Sigma + 1^1\Pi_{+1} + 2^1\Pi_{+1}$ superposition state driven by the coinciding laser pulses with equal (++) circular polarization, for a scenario with NaK's axis tilted by $\cos \theta = 0.7$ and an arbitrary value of φ . The resulting snapshots of the electron density are shown in the Figure. Obviously, they are similar to those shown in Figure 2, but less pronounced. The reduction of the overall amplitude depends on $\cos \theta$, irrespective of the value of φ .

Same as Fig. 2a but for NaK's axis tilted by $\cos \theta=0.7$ and an arbitrary value of φ .

In conclusion, the new quantum dynamics simulation for non-perfect orientation show that the impact of finite orientation (imperfect axis distribution) is to reduce the continuous chirality measure CCM which we have discussed in reply to Reviewer #2, Q2, but the essentials of the chirality flips e. g. the periods in femtosecond and attosecond time domains are maintained. The possibility to observe the electronic chirality flips is the same as discussed in reply to Reviewer #2, Q1, see also the caveat which is added below in reply to Reviewer #3, Q2. Accordingly, the impact on effects induced by electronic chirality flips, as proposed for the combination with Chiral Induces Spin Selectivity, will be maintained in reduced manner.

The newly added References are:

13. Kitano, K., Ishii, N. & Itatani, J. High degree of molecular orientation by a combination of THz and femtosecond laser pulses, *Phys. Rev. A* **84**, 053408 (2011).
14. Liu, J.-S., Cheng, Q.-Y., Yue, D.-G., Zhou, X.-C & Meng, Q.-T. Influence factor analysis of field-free molecular orientation, *Chin. Phys B* **27**, 033301 (2018)
15. Chordiya, K., Simkó', I., Szidarovszky, T. & Kahaly, M. U. Achieving high molecular alignment and orientation for CH₃F through manipulation of rotational states with varying optical and THz laser pulse parameters, *Sci. Rep.* **12**, 8280 (2022)
16. Grohmann, T. On the possibility of frozen nuclei. *Mol. Phys.* **119**, e1837405 (2021).
53. Altmann, S. L., *Rotations, Quaternions, and Double Groups*, (Clarendon Press, Oxford, 1986).
54. Varshalovich, D. A., Moskalev, A. N. Khersonskii, V. K. Quantum Theory of Angular Momentum (World Scientific, United States, 1987).

Modifications of the text:

1. In the 2nd paragraph of the Section for “The concept”, we have added in line 6:
“...for the molecule NaK which has been pre-oriented by z-polarized THz and femtosecond lasers pulses¹³⁻¹⁵; the details are in the Supplementary Information (SI). For reference, we shall assume perfect pre-orientation even though this cannot be achieved in practice¹⁶. The example NaK has already served to discover...”
2. In the last paragraph of the Conclusion Section, we have added in line 3 from bottom: “...time scale, analogous to the switches of the direction of spin-polarization induced by chirality flips during the rotation of a molecular motor⁵². The effects will be reduced by finite orientation (imperfect axis distribution).^{13-16, 53,54} By extrapolation ...”

2) The authors mention photoelectron circular dichroism as a possible way to detect the electronic chirality. However, as the authors know, PECD is often dominated by the chirality of the scattering potential, rather than the chirality of the electronic wavefunction. Comments on this aspect would be highly welcome.

Our response: We agree of course with Reviewer #3's proviso that PECD images of electronic wavefunctions or the corresponding electronic densities at time t are affected by the vector potential $A(t)$ of the photoionizing laser pulse, because the final electron momentum \mathbf{p} at the detector is approximately equal to the sum of the initial momentum $m_e\mathbf{v}$ of the photoelectron plus $A(t)$. If one aims at the reconstruction of three-dimensional electron densities, one must, therefore, eliminate the effects of $A(t)$. This remains a huge challenge not only for PECD but for all techniques of time-resolved imaging via photoionization. Nevertheless, for our purpose of monitoring the chirality flips, one can exploit the fact that opposite electronic enantiomers provide different initial momenta $m_e\mathbf{v}$. As a consequence, different enantiomers yield different images, irrespective of the effect of $A(t)$. Imaging by means of time resolved photoionization could, therefore, provide quantitative information about the periods of the present periodic chirality flips, and about the equivalence of the time-dependent electronic chirality flips in the femtosecond and attosecond time domains as generated by coinciding laser pulses with equal versus opposite circular polarizations, respectively. We conclude, therefore, that PECD or other methods based on time-resolved photoionization could still provide valuable information about the present chirality flips. Nevertheless, to avoid Reviewer#3's proviso about PECD, we now prefer it to recommend monitoring the chirality flips by means of time-resolved x-ray scattering³⁸⁻⁴⁰, as explained in our reply to item 1 of Reviewer #2. Accordingly, we deleted the hint to PECD in the Abstract and in the conclusions, and we have added Reviewer #3's proviso at the end of the new paragraph which we have added in reply to Reviewer #2.

Modifications of the text:

- (i) In the last sentence of the Abstract, we have deleted the previous hint to PECD. The modified last sentence of the Abstract thus reads: “As possible application, we propose the combination of the electronic chirality flips with Chiral Induced Spin Selectivity.”
- (ii) We have deleted the previous hint to PECD in the first and second sentences of the last paragraph of the conclusions. The new first sentence of the last paragraph of the conclusions

is essentially equal to the previous third sentence. It reads: “As possible application, we propose to combine the present electronic chirality flips with the effect of chiral induced spin selectivity (CISS).”

(iii) We have added Reviewer #3’s proviso at the end of the new paragraph which we have added in reply to item 1 of Reviewer #2. The complete newly added paragraph (with new References [38-46]) between the new modified last paragraph and the second last previous paragraph reads as follows:

“... in the electronic density.

Recent quantum dynamics simulations of the imaging of charge migration in chiral molecules suggest that the electronic chirality flips can be measured by time-resolved x-ray diffraction³⁸. The differential scattering probabilities (DSP) are calculated by consistent quantum electrodynamics description of light and quantum mechanical description of the electron dynamics³⁹. Different DSPs for opposite electronic enantiomers imply that the present periodic electronic chirality flips should yield periodically alternating DSPs. The required ultrashort x-ray pulses are available, cf. Ref. 40. Promising complementary techniques include chiral spectroscopy by dynamical symmetry breaking in high harmonic generation⁶, ultrafast imaging of chiral dynamics by non-linear enantio-sensitive optical response to synthetic chiral light⁴¹ ultrafast photoelectron imaging of attosecond electron dynamics^{5,42-45}, or photoelectron circular dichroism (PECD)^{7,10,37,46}. As a caveat, the reconstruction of three-dimensional time-dependent electron densities by means of the images obtained via photoionization, such as PECD, requires the elimination of the contributions caused by the electric fields which induce photoionization. Nevertheless, time-resolved variations of the images of alternating enantiomers could provide valuable quantitative information about the periods of the chirality flips and about their equivalence in the femtosecond and attosecond time domains, as generated by coinciding laser pulses with the same or with opposite circular polarizations, respectively.

As possible applications, we propose ...”

3) The authors make the point that a single circularly polarized laser pulse would not be capable of inducing the electronic chirality. I am not sure that I understand this aspect because a single ultrashort broadband pulse that covers the energies of both excited states and a controllable spectral phase should be capable of mimicking the effect of the two 8-fs pulses with different frequencies

Our response: *We appreciate it very much, and we are grateful to Reviewer #3 for addressing this stimulating aspect. Our response is divided into two parts:*

(i) On one hand, one can in fact replace the two coinciding resonant laser pulses with equal circular polarizations (++ or --) by a single ultrashort broadband transform-limited laser pulse with the same circular polarization (+ or -), in order to generate the superposition of the electronic ground state $1^1\Sigma^+$ plus two non-degenerate excited Π states with the same helicities ($1^1\Pi_{+1}$ and $2^1\Pi_{+1}$ or $1^1\Pi_{-1}$ and $2^1\Pi_{-1}$, respectively). Irrespective of the laser pulse(s), the resulting superposition states yield equivalent chiral electron densities, with the same periods

of the chirality flips in the femtosecond time domain. Indeed, we have verified the conjecture of Reviewer #3 by carrying out additional quantum dynamics simulations of the electronic wavefunction of oriented NaK driven by a single circularly polarized ultrashort broadband laser pulse with parameters $\tau = 4$ fs, $\omega = 3.784$ fs⁻¹, $\epsilon = 0.729$ GV/m corresponding to maximum intensity $I_{\max} = 0.14$ TW/cm². The effect of this single off-resonant laser pulse is equivalent to the previous two resonant laser pulses, i. e. they generate equivalent superposition states representing chiral electron densities with the same periods of chirality flips.

- (ii) On the other hand, it is impossible to replace the two coinciding resonant laser pulses with opposite circular polarizations (+ - or - +) by a single ultrashort broadband laser pulse to generate the superposition of the electronic ground state $1^1\Sigma^+$ plus two non-degenerate excited Π states with opposite helicities ($1^1\Pi_{+1}$ and $2^1\Pi_{-1}$ or $1^1\Pi_{-1}$ and $2^1\Pi_{+1}$, respectively). Hence, a single ultrashort broadband laser pulse cannot induce flips of chiral electron densities with periods in the attosecond time domain.

As resume, Reviewer #3's item 3) points to the property which really matters: It is the superposition of the electronic ground state $1^1\Sigma^+$ plus two non-degenerate excited Π states, either with the same or with opposite helicities for the representation of chiral electron densities with periods of the chirality flips in the femtosecond or attosecond time domains, respectively. It is important to design the laser pulses which generate these superposition states, but the specific realizations of the laser pulses are irrelevant.

Modifications of the text: We have modified the beginning of the third paragraph of the Conclusions such that it emphasizes the decisive role of the superposition states but eliminates the discussion of the irrelevant number of circularly polarized laser pulses which are necessary to generate the superposition states. Specifically, the new Beginning of the third paragraph of the Conclusions reads as follows:

“The present chirality of the electronic density is generated in a superposition state of three electronic eigenstates of the oriented NaK, namely the ground state and two excited states which have the same irreducible representations (here: Π) different from the ground state (Σ^+) (see eqn. (1)). The superposition of three eigenstates is unique. For comparison, a superposition of two eigenstates could not break all the symmetry elements of $C_{\infty v}$, which means the electron density would remain achiral. Superpositions of more than three non-degenerate eigenstates with incommensurable frequencies would not allow periodic electron dynamics. The present superposition of three molecular electronic eigenstates is reminiscent of the superposition of three atomic orbitals representing the chiral hydrogen atom^{7,37}. The phenomenon of periodic chirality flips is of course not restricted to the electronic density of oriented heteronuclear diatomic molecules. It should also exist in ...”

In summary, I strongly recommend this visionary work for publication in Nature Communications, with optional revisions as described above.

Our response: Again, we are grateful to Reviewer #3, and we are of course glad about the positive evaluation.

REVIEWERS' COMMENTS

Reviewer #1 (Remarks to the Author):

My concerns have been addressed in the revised version.

I believe the paper is suitable for publication in "Nature Communications".

Reviewer #3 (Remarks to the Author):

The revised manuscript adequately resolves all the points raised by the 3 reviewers. I strongly recommend publication of this visionary work.